# Bi-Resolution Hash Encoding in Neural Radiance Fields: A Method for Accelerated Pose Optimization and Enhanced Reconstruction Efficiency

**Zixuan Guo** [1,2] 🔵, **Qing Xie** [1,2], **Song Liu** [1] **and Xiaoyao Xie** [1,*]

1   Guizhou Key Laboratory of Information and Computing Science, Guizhou Normal University, Guiyang 550001, China; 20030060024@gznu.edu.cn or math@gznu.edu.cn (Z.G.); 19030060023@gznu.edu.cn (Q.X.); songliu@gznu.edu.cn (S.L.)
2   School of Mathematical Science, Guizhou Normal University, Guiyang 550001, China
*   Correspondence: xyx@gznu.edu.cn

**Featured Application: This research addresses an essential component in the practical application of NeRF by working toward enhancing the reconstruction efficiency, especially in scenarios without precise pose estimation. The improvements presented in this research underscore its growing potential for real-world deployment. Potential application areas include scenes wherein feature extraction is challenging, such as ceramics from archaeological excavations that are smooth and low-textured, artifacts with repetitive textures, certain repetitive terrains and landforms in Geographic Information Systems (GIS) and remote sensing, among others.**

**Abstract:** NeRF has garnered extensive attention from researchers due to its impressive performance in three-dimensional scene reconstruction and realistic rendering. It is perceived as a potential pivotal technology for scene reconstruction in fields such as virtual reality and augmented reality. However, most NeRF-related research and applications heavily rely on precise pose data. The challenge of effectively reconstructing scenes in situations with inaccurate or missing pose data remains pressing. To address this issue, we examine the relationship between different resolution encodings and pose estimation and introduce BiResNeRF, a scene reconstruction method based on both low and high-resolution hash encoding modules, accompanied by a two-stage training strategy. The training strategy includes setting different learning rates and sampling strategies for different stages, designing stage transition signals, and implementing a smooth warm-up learning rate scheduling strategy after the phase transition. The experimental results indicate that our method not only ensures high synthesis quality but also reduces training time. Compared to other algorithms that jointly optimize pose, our training process is sped up by at least 1.3x. In conclusion, our approach efficiently reconstructs scenes under inaccurate poses and offers fresh perspectives and methodologies for pose optimization research in NeRF.

**Keywords:** NeRF; multi-resolution hash encoding; pose optimization; reconstruction efficiency

## 1. Introduction

In computer vision, the advent of NeRF (neural radiance fields) technology has enabled photo-realistic rendering, greatly advancing the development of more lifelike virtual environments. This technology shows great potential in various fields, including urban digitization [1,2], autonomous driving [3,4] and the creation of virtual characters [5–7]. Particularly in virtual reality (VR), the application of NeRF technology has enhanced the realism of scenes, providing users with a deeper immersive experience. Research combining NeRF and VR technologies continues to progress. For instance, FoV-NeRF [8] significantly reduces latency by incorporating eye-tracking technology, improving perceptual quality in environments with high fields of view and resolution. The Re-ReND [9] method optimizes

the rendering of neural radiance fields in VR devices, achieving high efficiency and minimal quality loss. Additionally, VR technology has extended into fields such as medicine, education and psychology. For example, the application of VR and AR in ophthalmic diagnosis and screening has shown their potential in disease diagnostics [10]. These technologies also play a crucial role in professional skills training, for example, in the training of nursing students [11], otolaryngology surgeons [12] and adults with autism spectrum disorders [13], where VR significantly enhances participants' confidence levels. In conclusion, with the ongoing development and application of NeRF technology, work related to more immersive scenes is steadily improving, revealing its vast potential and value in engineering science. In the NeRF-based method, the photo-realistic rendering heavily relies on precise 3D scene reconstruction. However, the information usually recorded in the photographing process is the scene's photo-metric information, not the camera's pose, which means that it does not include the specific location and orientation of the camera when each photo is taken. Therefore, further estimation of the camera pose is required. Currently, there are two main methods for estimating camera pose in NeRF: the traditional method based on feature point matching and the joint pose optimization method based on photo-metric information between images. Nevertheless, these two methods have the following problems:

1.  The traditional method of camera pose estimation relies on detecting and matching feature points. However, this method encounters significant difficulties in scenes lacking obvious features [14], such as low textures or reflections. The lack of sufficient feature points in these scenes makes it difficult for traditional algorithms to accurately calculate the camera pose, thereby affecting the quality of the entire 3D scene reconstruction.
2.  Although the joint pose optimization method can have better effects on areas with low texture, this method is currently mainly based on the Multilayer Perceptron (MLP) model, which is slow to converge and requires more processing time and resources. This poses a significant limitation for the research and application of such algorithms.

To address the aforementioned issues, this paper introduces multi-resolution hash encoding within the framework of the joint pose optimization method. This improved approach not only has more significant advantages in scenarios with low textures and reflections but also allows for more efficient pose adjustment and reduced reconstruction time. Specifically, this paper proposes a scene reconstruction method named BiResNeRF, which employs a neural network architecture with a fusion module of hash encoding at two different resolutions. Furthermore, a two-stage training strategy is adopted based on this architecture. In the strategy, low-resolution and high-resolution hash encoding modules are, respectively, responsible for pose estimation and high-precision scene reconstruction during the training. This method not only ensures the quality of the scene reconstruction but also achieves a more efficient reconstruction process.

The main contributions of this research work are as follows:

1.  We propose a method for scene reconstruction called BiResNeRF, which is based on low and high-resolution hash encoding modules. The proposed approach ensures rapid and accurate scene reconstruction even in the presence of inaccurate poses.
2.  We delve deeply into the relationship between hash encoding of varying resolutions and pose estimation. Through experimental analysis, the characteristics of pose estimation related to both low and high-resolution hash encodings were explored, providing theoretical and empirical foundations for the research presented in this paper as well as for subsequent works.
3.  To ensure the effective training of BiResNeRF, a two-stage training strategy is proposed, with the transition between stages being timely and completed by the real-time detection of error stability signals. At the same time, a coarse-to-fine sampling strategy and a smooth warm-start learning rate scheduling strategy are adopted to make the training process proceed smoothly and efficiently.
4.  The effectiveness of our algorithm has been experimentally verified when applied to scenes with characteristics such as the absence of pose, low texture and reflection.

This demonstrates the application value of the algorithm and its potential importance in future research.

In this paper, Section 2 provides a brief overview of related work. In Section 3, we first present the mathematical formulation, followed by an introduction to the neural network framework and the corresponding training strategies we used. Section 4 comprehensively validates our methods, including theoretical aspects, comparative experiments related to our method and a reconstruction experiment in a low-texture scenario. Section 5 discusses the findings and implications of our study, highlighting some limitations of our method and potential directions for future work. Finally, we summarized the main findings and conclusions of this paper in Section 6.

## 2. Related Work

In this paper, we primarily focus on two aspects of the neural radiance field: pose estimation and reconstruction efficiency. Subsequent sections organize and compare the relevant work related to these two aspects.

### *2.1. Pose Optimization Related*

Based on whether they rely on feature point extraction and matching to calculate pose data, pose estimation methods can be divided into feature-based methods and joint pose optimization methods.

### 2.1.1. Feature-Based Methods

Feature-based methods typically utilize epipolar geometry to determine the pose of a camera, frequently employed in SFM (Structure from Motion) and SLAM (Simultaneous Localization And Mapping) systems. For instance, in SLAM systems, ORB-SLAM [15] utilizes ORB feature points for pose estimation, boasting rapid and robust characteristics, extensively adopted in the robotics domain. For tasks demanding high surface precision, SIFT [16] is typically employed for feature extraction due to its superior matching accuracy. In implicit scene reconstruction, the dataset used in the NeRF [17] paper can apply the aforementioned methods in SFM and SLAM to compute feature points, subsequently determining the pose data. Additionally, COLMAP [18] proposed a robust 3D reconstruction system and offered an open-source tool. This tool facilitates feature point extraction and camera pose calculation in scenes, providing convenient conditions for NeRF researchers. Subsequent works based on NeRF [19–21] all employ feature-based methods. Furthermore, the SaNeRF [22] algorithm incorporates epipolar constraints to achieve 3D scene reconstruction without pose data, but this method still relies on SIFT feature point matching.

### 2.1.2. Joint Pose Optimization Methods

Joint pose optimization refers to the methodology that does not rely on feature points and optimizes the camera pose during the neural network training process. NeRF− [23] introduced a framework that jointly optimizes camera parameters and neural radiance field parameters, enabling scene reconstruction in situations with camera disturbances or unknown camera parameters. Around the same time, BARF [24] adopted a framework similar to NeRF– and theoretically linked it to 2D image registration, adding the theoretical understanding for joint pose optimization. The authors also found that high-frequency pose data impede pose training. Hence, a coarse-to-fine training strategy was adopted. GARF [25] presented a Gaussian activation neural radiance field, enabling 3D scene reconstruction without utilizing positional encoding. We also demonstrated the superiority of direct methods when handling low-texture areas. GNeRF [26] introduced a method combined with GAN to estimate camera pose under conditions wherein the camera pose is entirely unknown and scene conditions may be complex. However, this approach relies on a camera sampling distribution not far from the true distribution to prevent training failures. The research on the mentioned methods has contributed to the methods of joint

pose optimization. However, challenges persist, such as the high demands for initial pose estimation and the relatively low efficiency of reconstruction, which significantly hinder the advancement of research, impede development efforts and limit the practical application of related algorithms.

### 2.2. Reconstruction Time-Related

NeRF can generate high-resolution images with a small spatial footprint. However, the original NeRF used an MLP-based neural network representation, resulting in slow reconstruction and inference speeds, which is one of the limiting factors in its practical application. Several methods have emerged in recent years aimed at improving NeRF's reconstruction efficiency.

EfficientNeRF [27] employs effective sampling and a new data structure to accelerate NeRF's training and inference processes, reducing training time by more than 88%. However, it still requires several hours of training. Plenoxels [28] adopts an approach based on sparse voxel grids and spherical harmonics, eliminating the need for neural networks. This method achieves faster training speeds than the baseline NeRF model with comparable quality. However, it necessitates manual parameter adjustments depending on the specific scene. Both DVGO [29] and TensorRF [30] optimize directly on voxel grids. DVGO, which optimizes voxel grids directly, achieves a training speed 45 times faster than the baseline NeRF but struggles with scenes without boundaries or forward-facing scenes. TensorRF employs various tensor decomposition techniques to reduce memory usage and achieve efficient rendering, though it only supports bounded scenes. Moreover, Instant-ngp [31] utilizes multi-resolution hash encoding techniques, significantly reducing the computational cost of representing high-resolution image features and cutting training times down to just a few seconds from minutes. These methods have played a vital role in advancing implicit representation reconstruction. However, they all rely on precise pose information.

Furthermore, some research focuses on improving reconstruction efficiency based on joint pose optimization methods. NeRFAcc [32] builds on BARF, introducing an occupancy grid that accelerates BARF by skipping empty areas. In research concurrent with our method, BAA-NGP [33] attempts to integrate multi-resolution hash encoding into the joint optimization framework to speed up scene convergence. However, BAA-NGP needs to clearly elucidate the intrinsic relationship between multi-resolution hash encoding and pose estimation, necessitating further discussion.

In summary, studies on joint pose optimization still suffer from the problem of low reconstruction efficiency. To address this, we have incorporated multi-resolution hash encoding into the joint pose optimization method and explored the relationship between pose estimation and hash encoding at different resolutions. A series of subsequent experiments have provided theoretical and experimental evidence for the improved approach presented in this paper.

## 3. Methods

In Section 3.1, we first formulate a detailed description of the problems and objectives involved in this study. Subsequently, in Section 3.2, we elaborate on the neural network architecture of the proposed BiResNeRF method. Following this, in Section 3.3, we discuss the corresponding training strategy.

### 3.1. Formulation

In NeRF, it is assumed that the scene is composed of luminous particles with a density that can change. The magnitude of the density determines whether there is an object at a given position for the ray. The density field is denoted by $\sigma(\mathbf{x})$, where $\mathbf{x} \in \mathbb{R}^3$ indicates the distribution of density in space. We re-parameterize $\mathbf{x}$ using the center of the camera and the pixels on the imaging plane to determine the origin $\mathbf{o}$ and direction $\mathbf{d}$ of the ray. Any point $\mathbf{x}$ along the ray can be represented as $\mathbf{r}(t) = \mathbf{o} + t\mathbf{d}$. The color and density of the

scene can be represented by the continuous function $F_\theta$, where $\theta$ denotes the parameters of the neural network.

$$F_\theta : (\mathbf{r}(t), \mathbf{d}) \to (\mathbf{c}, \sigma) \tag{1}$$

where $\mathbf{c}$ is the color vector and $\sigma$ is the scalar for density. The choice of mapping method for the neural network directly affects its efficiency and output quality. In NeRF, there are mainly two ways to map $F_\theta$. One is the neural network mapping based on MLP, which has the advantage of occupying less storage space but has a slower convergence speed. The process for this mapping can be detailed in the following formula.

$$(\sigma, \mathbf{c}) = MLP_{out}(MLP_{pos}(PositionEncoding(\mathbf{x})), DirectionEncoding(\mathbf{d}))) \tag{2}$$

$MLP_{pos}$ is a multi-layer perceptron network that encodes positional information, while $MLP_{out}$ is another MLP tasked with outputting color and density.

Another is the neural network mapping based on multi-resolution hash encoding. It has the advantage of converging while maintaining high accuracy. The process for this mapping can also be represented by the subsequent formula.

$$(\sigma, \mathbf{c}) = MLP_{out}(HashGrid(\mathbf{x}), DirectionEncoding(\mathbf{d})) \tag{3}$$

The final color is determined by the volume rendering formula:

$$C(\mathbf{r}) = \int_{near}^{far} T(t) \cdot \sigma(t) \cdot \mathbf{c}(t) dt \tag{4}$$

$C(\mathbf{r})$ is the color observed along the ray $\mathbf{r}$. $T(t)$ is the transmittance accumulation function, and the range of integration is from *near* to *far*, representing the interval along the ray from the nearest point to the farthest point.

Given a sequence of images denoted by $\{I_i\}_{i=1}^m$, $m$ is the total number of images and $i$ is the index of the image. The origin o and direction $\mathbf{d}$ are determined by the camera pose $\mathbf{p}$ and pixel position $u$. Specifically, the origin $\mathbf{o}$ is primarily determined by the camera pose and can be represented as $\{\mathbf{o}(\mathbf{p_i})\}_{i=1}^m$. The direction $\mathbf{d}$ is determined by both $\mathbf{p}$ and $u$, and can be represented as $\{\mathbf{d}(\mathbf{p_i}, u_j)\}_{i=1,j=2}^{m,n}$, where $n$ is the number of pixels in an image and $j$ is the index of the pixel.

Finally, gradient descent and backpropagation are used to minimize the loss function and optimize the camera and neural radiance field parameters. The formula is presented below. This expression denotes the estimated pixel value, while $l$ can assume values of 1 or 2, corresponding to the $L_1$ and $L_2$ norms, respectively. $\hat{I}$ represents the estimated value of the pixel with index $u_j$ in the $i$-th image.

$$\min_{\mathbf{p_1}, \cdots, \mathbf{p_m}, \theta} \sum_{i=1}^m \sum_{j=1}^n ||\hat{I}(u_j; \mathbf{p_i}, \theta) - I_i(u_j)||_l \tag{5}$$

### 3.2. Neural Network Architecture of BiResNeRF

The network used in this paper adopts a framework similar to the joint pose optimization and neural radiance field in BARF [24] and NeRF− [23]. The overall network architecture is shown in Figure 1. BiResNeRF is an improvement on the neural radiance field model part of NeRF based on this framework. The improved network architecture is shown in Figure 2.

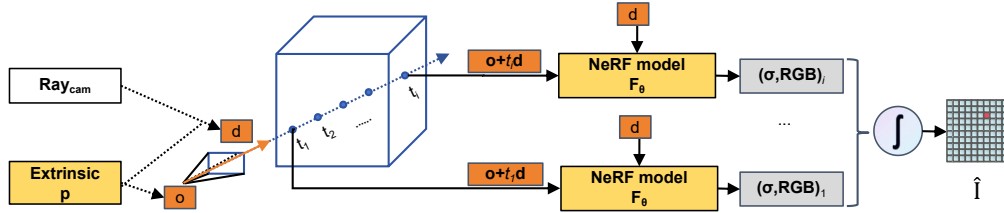

**Figure 1.** The framework of joint pose optimization.

As shown in Figure 1, the yellow parts represent the training parameters, including $\{\mathbf{p}, F_\theta\}$. $\mathbf{o}$ and $\mathbf{d}$ are the origin and direction in the world coordinate system of the ray, calculated based on $\mathbf{Ray_{cam}}$ and the extrinsic parameter $p$. Sampling is performed on this ray, and the position of the sample point can be parameterized as $\mathbf{o} + t_i\mathbf{d}$, where $t_i$ represents the depth of the $i$th sample point. The position of the sample point and its direction $\mathbf{d}$ are input into the neural radiance field model, which returns the density and color of this point. Finally, by integrating the sampled points along the ray, the final color of the ray is obtained.

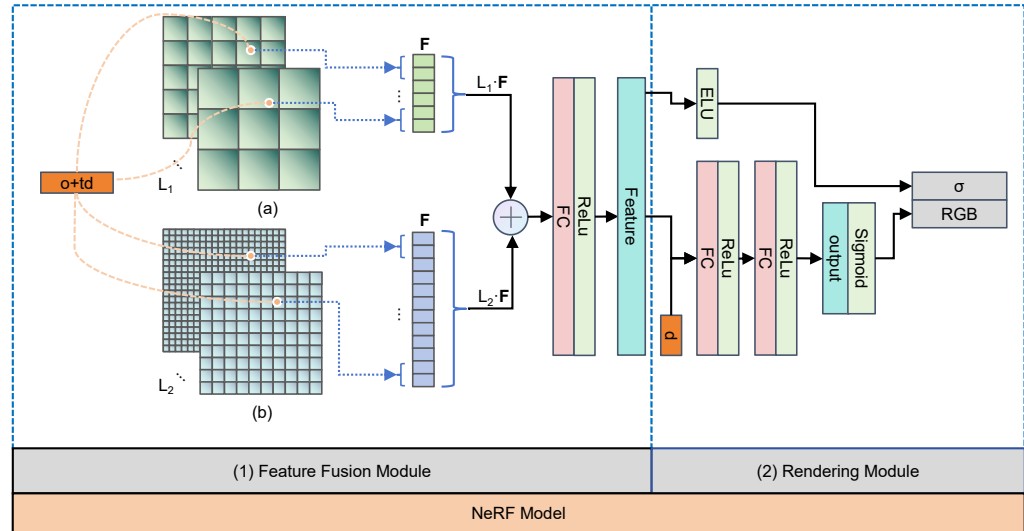

**Figure 2.** NeRF model network architecture.

As shown in Figure 1, this architecture consists of two modules: (1) the feature fusion module and (2) the rendering module. The feature fusion module is composed of two submodules, namely the low-resolution pose estimation module (a) and the high-resolution reconstruction module (b). These two modules have L1 and L2 different resolution layers, respectively. Each resolution layer obtains a feature vector of length F, and after concatenation, the length of the feature vector is (L1 + L2) × F. Finally, this module uses a multi-layer perceptron (MLP) with one hidden layer to perform feature fusion and obtain the final feature vector. The primary function of the rendering module is to extract the density and color of the given position. The first component of the feature vector obtains the density $\sigma$ through the ELU activation function. The other components of the feature vector are concatenated with the ray direction as new input features. Lastly, they pass through an MLP with two hidden layers, each having 64 neurons, and the Sigmoid activation function to obtain the RGB value of that position.

When multi-resolution hash encoding is directly introduced in scenes with inaccurate poses, the features of a given position are generated under the combined effect of hash encodings of different resolutions. This hinders the pose estimation task due to the high-resolution hash encoding, making it difficult to adjust effectively, thereby affecting the accurate reconstruction of the scene.

We propose the Low-Resolution and High-Resolution Hash Encoding feature fusion module, abbreviated as the feature fusion module, to address the issue of ineffective pose adjustment. As shown in Figure 2, this module consists of two sub-modules: the Low-Resolution pose estimation module (a) and the High-Resolution reconstruction module (b). The pose estimation module has fewer layers of low-resolution hash encoding. The larger grid provides more space for pose adjustment, enabling better completion of pose estimation tasks. Module (b) includes more layers of high-resolution hash encoding. The smaller grid has lesser pose adjustment capabilities but excels at capturing details, resulting in better high-resolution reconstruction.

### 3.3. Two-Stage Training Strategy

Section 3.3 elaborates on the proposed two-stage training strategy based on the architecture. The strategy is divided into four key components: an overview of the two-stage training strategy, real-time error stability detection method, coarse-to-fine ray sampling strategy, and smooth warm-up learning rate scheduling strategy.

### 3.3.1. Overview

Due to the limitations brought about by the high-resolution hash encoding module on the performance of pose estimation tasks, we adopt a two-stage training strategy to achieve stable and efficient scene reconstruction. The process of training is shown below.

As shown in Figure 3, the main task of the first stage is pose estimation. At the beginning of training, a higher learning rate must be set for the pose estimation module in the feature fusion module to achieve faster adjustment on pose parameters. At the same time, the high-resolution reconstruction module is set and maintained at a low learning rate to restrict its convergence. During the training process of this stage, it is also necessary to determine whether the pose error is stable. If it is unstable, the training continues; if it is stable, the second stage is entered. In the second stage, the pose estimation module maintains the original learning rate, while the high-resolution reconstruction module uses a smooth warm-up learning rate scheduling strategy to ensure a smoother and faster scene reconstruction process. Next, the model continues to train until the convergence condition is met or the predetermined maximum number of iterations is reached.

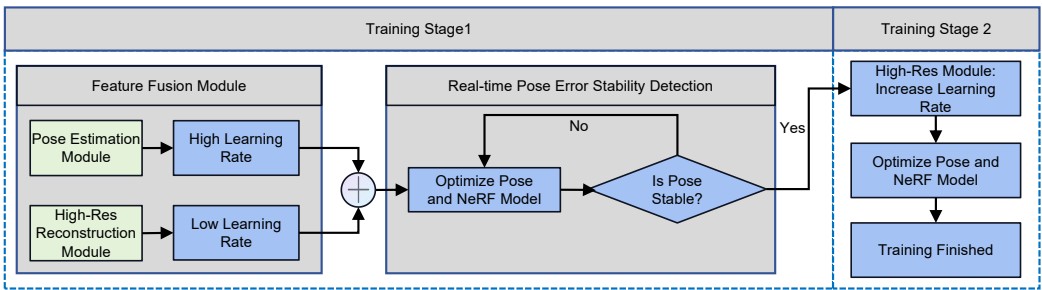

**Figure 3.** Two-stage training flowchart.

In the following, we provide a detailed explanation of the key components of the training strategy.

### 3.3.2. Real-Time Error Stability Detection Method

Determining how to transition in a timely manner from the first stage of training to the second stage, in order to enhance the overall training efficiency, is a key issue addressed during the training process. For this purpose, we introduces a real-time error stability detection method.

Figure 4a,b illustrate two common trends in error variation: decrease and increase. The red points, $center_1$ and $center_2$, represent the average values of errors in the first and second halves, respectively, while $\triangle E$ represents the absolute difference between the average values of the first and second halves. Figure 4c presents a schematic diagram of parameters

related to the degree of fluctuation. The red points still represent the averages in the first and second halves, while the red points represent the maximum and minimum errors in the first or second half. $\triangle E_1$ and $\triangle E_2$, respectively, indicate the differences between the maximum and minimum values in the first and second halves.

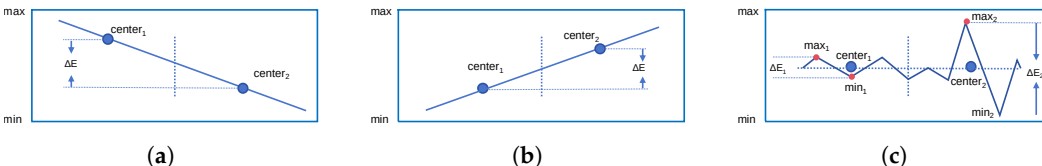

|     (a)     |     (b)     |     (c)     |

**Figure 4.** Schematic representation of the trend and degree of error variation. (**a**) Trend of error increase. (**b**) Trend of error decrease. (**c**) Degree of error fluctuation.

This method mainly relies on two key indicators: the trend of error variation and the degree of error fluctuation. When the downward trend of the error begins to slow (that is, the slope approaches zero) or even reverses into an upward trend, and the fluctuation ranges of the two subsequences are similar, it is determined that the error has reached a relatively stable state. At this point, the system emits a stage transition signal. In this paper, "error" refers to the value of the loss function. The specific algorithmic process is shown in Algorithm 1. In this algorithm, $E$ represents the error time series. $m_{prev}$ is the slope calculated from the previous measurement. $\tau_f$ is the fluctuation threshold we set, with a default value of 0.8, and $\tau_m$ is the threshold for the slope, with a default value of $-0.1$.

---

**Algorithm 1** Error Stability Signal Detection.

---

1: **function** STABILITYDETECTION($E, m_{\text{prev}}, \tau_f, \tau_m$)
2:     $n \leftarrow \text{length}(E)$
3:     $\Delta E \leftarrow \max(E) - \min(E)$
4:     $E_1 \leftarrow E[1 \ldots \frac{n}{2}]$
5:     $E_2 \leftarrow E[\frac{n}{2} + 1 \ldots n]$
6:     $Center_1 \leftarrow \frac{2}{n} \sum_{i=1}^{\frac{n}{2}} E_1[i]$
7:     $Center_2 \leftarrow \frac{2}{n} \sum_{i=\frac{n}{2}+1}^{n} E_2[i]$
8:     $m \leftarrow \frac{Center_2 - Center_1}{\Delta E}$
9:     $\Delta E_1 \leftarrow \max(E_1) - \min(E_1)$
10:     $\Delta E_2 \leftarrow \max(E_2) - \min(E_2)$
11:     $F \leftarrow \frac{\min(\Delta E_1, \Delta E_2)}{\max(\Delta E_1, \Delta E_2)}$
12:     **if** $m > \tau_m$ **and** $F > \tau_f$ **then**
13:         $S \leftarrow \text{True}$
14:     **else**
15:         $S \leftarrow \text{False}$
16:     **end if**
17:     **return** $m, S$
18: **end function**

---

When setting $\tau_m$, it should be noted that the closer the value is to 0, the smaller the change in the loss function, indicating a potential convergence. At this point, a stage transition may be considered. The value is the main factor affecting trend determination, and its range should not be too large. The range between $-0.3$ and $-0.1$ can be considered.

When setting $\tau_f$, consider that the training process is a continuous optimization, with the loss function consistently undergoing increases and decreases, resulting in instability. A too-high threshold makes the conditions overly strict, hindering stage transitions. On the other hand, a too-low threshold may misjudge phases of rapid change. The closer this value is to 1, the more similar the fluctuation levels of the two subsequences, and the smaller the value, the greater the fluctuation of the loss function, indicating an unstable training process. A range between 0.4 and 0.8 can be considered.

### 3.3.3. Coarse-to-Fine Ray Sampling Strategy

To achieve faster pose convergence and higher rendering quality, we need to set different sampling intervals along the rays for different stages.

In the training of BiResNeRF, both the batch size of rays and the sampling strategy on the rays are factors that affect the convergence speed of the model. Due to constraints on computational resources, when the total number of sampled points is fixed, we can handle larger batches of rays by reducing the number of sampled points on each ray. As a result, when more rays are involved in the training, this means that more scene information can be obtained, leading to a more effective learning of the pose estimation and scene representation. However, due to the decrease in the number of sampled points, fewer points on each ray will participate in the volume rendering process, inevitably leading to a lower rendering quality.

The two-stage coarse-to-fine sampling strategy effectively speeds up the convergence rate of pose estimation in the initial stage, while enhancing the quality of the scene in the later stage.

### 3.3.4. Smooth Warm-Up Learning Rate Scheduling Strategy

When transitioning to the next stage, to ensure a smooth training process, we need to use a smooth warm-up learning rate scheduling strategy.

In the first stage of training, the poses after full adjustment usually approach their true states, and significant changes in subsequent adjustments are not desirable. A sudden increase in the learning rate can disrupt the optimization process, potentially leading to significant deviations in some poses impacting the subsequent reconstruction work. The learning strategy we propose allows for smooth adjustments of the learning rate, both when increasing and decreasing, as illustrated in the following formula:

$$
lr(i) = \begin{cases} lr_0 + (lr_{\max} - lr_0) \cdot G(i) & \text{if } 0 \leq i < I_{\text{inc}} \\ \max(lr_{\max} \cdot D(i), lr_{\min}) & \text{if } I_{\text{inc}} \leq i < I_{\text{total}} \\ lr_{\min} & \text{if } i \geq I_{\text{total}} \end{cases} \tag{6}
$$

where $lr_0$ is the initial learning rate, $lr_{max}$ is the maximum value of the learning rate after warm-up, $lr_{min}$ is the minimum value for learning rate decay, $i$ is the current iteration round, $I_{total}$ is the total number of iterations and $I_{inc}$ is the number of rounds for learning rate increase. $G(i)$ is the growth factor:

$$
G(i) = \frac{1}{2} \cdot \left( \sin \left( -\frac{\pi}{2} + \pi \cdot \frac{i}{I_{\text{inc}}} \right) + 1 \right) \tag{7}
$$

$D(i)$ is the decay factor:

$$
D(i) = \frac{1}{2} \cdot \left( 1 + \cos \left( \pi \cdot \frac{i - I_{\text{inc}}}{I_{\text{total}} - I_{\text{inc}}} \right) \right) \tag{8}
$$

In summary, an accurate pose is the basis for NeRF to achieve high-quality reconstruction of real scenes. In the case of joint optimization, the pose and neural radiation fields are optimized synchronously. During this process, the fluctuation of the pose directly affects the precision of the reconstruction. A solid foundation can be laid for the reconstruction details by rapidly adjusting the pose in advance, allowing for smooth progress and achieving higher precision.

## 4. Experiments

In this section, we mainly validate the effectiveness of our proposed method through three experiments. Firstly, we investigate the impact of different resolution hash encodings on pose estimation and verify the effectiveness of low-resolution grids for pose estimation.

Secondly, we analyzed the impact of the number of layers in the low-resolution module on pose estimation to further understand its architecture and performance. Thirdly, based on the aforementioned experiments, we experimentally verify the performance of BiResNeRF. Lastly, we conducted an experiment to validate the adaptability and effectiveness of our method when faced with scenarios characterized by the absence of pose, low texture and reflection.

**Dataset**

Experiments in this paper were conducted on the Realistic Synthetic 360° dataset [17]. This dataset, synthesized via Blender, encompasses eight objects with intricate geometric shapes and authentic non-Lambertian materials, namely Chair, Drums, Ficus, Hotdog, Lego, Materials, Mic and Ship. Images were generated by sampling and rendering as the camera orbited around these objects. Each object is represented with 100 training images and 200 test images, all accompanied by accurate pose data and rendered at a resolution of [800, 800] pixels. For the purposes of our experiment, the image size is reduced to half, resulting in a resolution of [400, 400] pixels.

**Evaluation Criteria**

We primarily evaluate the synthesis quality of new viewpoints and the accuracy of pose estimation. For the synthesis of new viewpoints, the Peak Signal-to-Noise Ratio (PSNR), Structural Similarity Index (SSIM) and Learned Perceptual Image Patch Similarity (LPIPS) [34] are employed as assessment metrics. These are commonly used image quality assessment indicators, capable of comprehensively reflecting the image's performance in pixel differences, structural information, and perceptual quality. For pose estimation, the Procrustes analysis [24] is used to align the estimated pose with the real pose, and subsequently, the rotation and translation errors are calculated.

### 4.1. The Impact of Resolution on Pose Estimation

#### 4.1.1. Experimental Setup

The experiment was evaluated under the Lego scene. To simulate situations with inaccurate poses, we introduced Gaussian noise N(0,0.15I) to the ground truth pose data, using it as the initial pose. The evaluation metrics adopted in this paper are rotation error, translation error and the PSNR. The computations were carried out on Ubuntu 20.04 within the Windows Subsystem for Linux (WSL), leveraging an Intel Core i7-13700K CPU and an NVIDIA GeForce RTX 3080 Ti GPU.

#### 4.1.2. Implementation Details

This experiment aims to explore the impact of resolution on pose optimization. The experiment adopts a joint pose optimization framework similar to BARF and uses single-resolution hash encoding to extract features from positions. The resolution size of the single-resolution hash encoding is set to 16, 32, 64, 96, 128, 258, 1024, with the number of layers set to 3, aiming to extract features more fully at this resolution. At the same time, we use the Adam optimizer, applying different learning rates and learning rate scheduling strategies for pose parameters and neural radiance field parameters. For the optimization of pose parameters, an exponential decay strategy is used with a decay coefficient of 0.99988, and the learning rate decays from $1 \times 10^{-3}$ to $1 \times 10^{-5}$. For the optimization of neural radiance field parameters, the initial learning rate is set to $1 \times 10^{-2}$, and a cosine annealing learning rate adjustment strategy is adopted, with the length of the learning rate annealing cycle set to 40,000. The entire training process was carried out for 40,000 iterations.

#### 4.1.3. Results of Different Resolution Hash Encoding

Figure 5a,b illustrate the trend of pose errors during the training process under different resolutions. As can be seen from the figure, at resolutions 16, 32 and 64, the errors are small and close to the real pose. When the resolution increases, the error becomes larger. At resolutions 96 and 128, it can be considered that a few cameras have pose deviations. At

resolutions of 256 and 1024, the error in pose is too large, suggesting that the majority of cameras have pose deviations and cannot complete the pose adjustment task. Figure 5c shows the trend of PSNR during the training process. Among them, the accuracy is the highest at a resolution of 96 and the lowest at resolutions 256 and 1024.

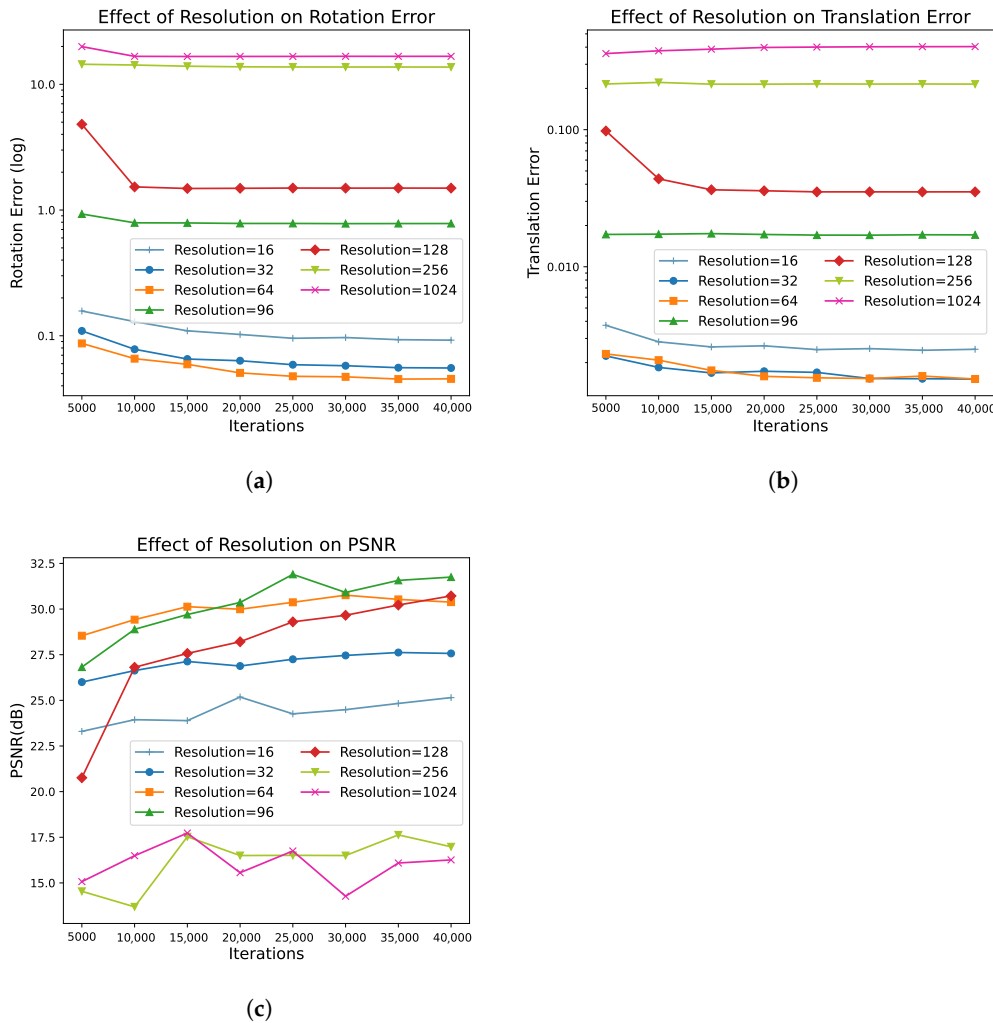

**Figure 5.** The impact of different resolutions on pose and rendering quality.

Through experiments, we have verified that the network based on hash encoding has the capability for pose estimation and rapid scene reconstruction. Additionally, it was validated that low-resolution hash encoding networks can rapidly and effectively perform pose estimation, whereas high-resolution hash encoding cannot function properly in this task. Moreover, we found that the highest rendering accuracy of the image is not the one with the smallest pose error. As shown in Figure 5, the rendering accuracy is the highest at a resolution of 96, but its pose estimation is not the most accurate. Accurate pose estimation is the foundation for high-precision rendering, so we cannot solely rely on rendering accuracy to judge the quality of the final result.

### 4.2. The Impact of Resolution Layers on Pose Estimation

The experiment used the same dataset and the same hardware and software configurations as in Section 4.1, but the experimental details differ from those in Section 4.1.

#### 4.2.1. Implementation Details

This experiment aims explore the impact of resolution layers on pose estimation. As described in Section 4.1, when the resolution is at 64, the result of pose estimation is close

to the true pose, but when it reaches 96, a small number of cameras have pose deviations. Therefore, this experiment chose the minimum resolution of 16 and the maximum of 72 and ensured that, after pose convergence, it could approach the real pose. The layer numbers chosen are 2, 3, 6, 8 and 16 to observe the effects of different numbers of layers on pose estimation. The resolution of each layer was calculated based on Instant-NGP [31].

For the pose parameters, the initial learning rate is set to $1 \times 10^{-3}$, with an exponential decay strategy and a decay coefficient of 0.99988. Regarding the neural radiance field parameters, the starting learning rate is determined as $1 \times 10^{-2}$, using a cosine annealing learning rate decay strategy, where the duration of the learning rate annealing cycle is set to 40,000 iterations. In the first experiment, the training process takes 20,000 iterations.

### 4.2.2. Results of Different Levels of Hash Encoding

As shown in Figure 6a,b, as the number of layers increases, the estimation error shows a decreasing trend. On the surface, this seems to suggest the use of more layers. However, a deeper analysis reveals a more complex picture. When further analyzing the PSNR values in Figure 6c, we find that the improvement in rendering accuracy is closely related to the accuracy of pose estimation. Specifically, the increased layers enhance the model's ability to capture hash-encoded features in the scene, resulting in a finer presentation of image details. As the image details continue to refine, the accuracy of the pose estimation increases, forming a mutually promoting cycle. However, what we really care about during the pose estimation phase is the convergence properties of the model, not the absolute accuracy. As illustrated in Figure 6a,b, starting from the second layer, the pose estimation trends of different layers are roughly the same, with no significant differences. This means that, with only two or three layers, we can achieve a stable estimation of the pose. Further increasing the number of layers will lead to an increase in model parameters, thus increasing the computational cost of training and inference.

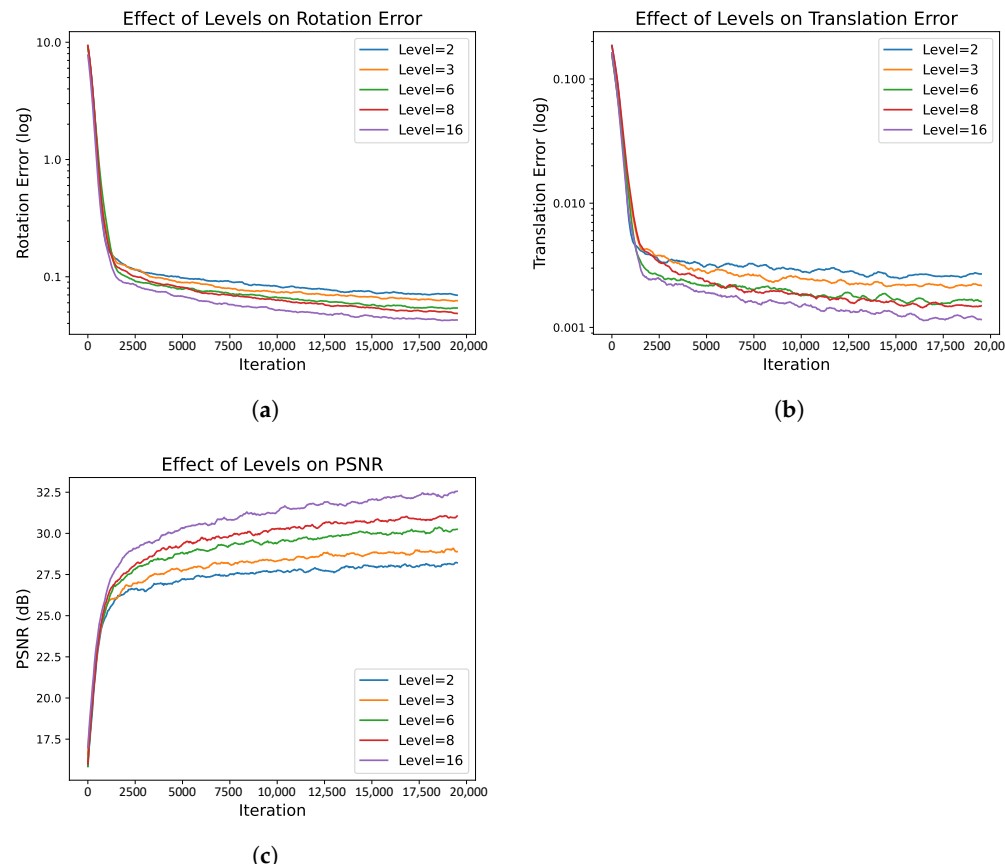

**Figure 6.** The impact of resolution layers on pose and rendering quality.

### 4.3. Experimental and Performance Analysis of BiResNeRF

#### 4.3.1. Experimental Setup

The experiment evaluated all scenes in the dataset. Evaluation criteria are rotational error, translational error, PSNR, SSIM, LPIPS and training time. The perturbation of the pose and the hardware and software environment of the experiment are the same as in previous experiments.

#### 4.3.2. Implementation Details

The architecture of BiResNeRF is shown in Section 3.2, and the specific parameters of the network structure are shown in Table 1. We use the Adam optimizer and adopt different learning rates and learning rate scheduling strategies for pose parameters and neural radiance field parameters. In the first stage for pose parameters, the learning rate decays from $1 \times 10^{-2}$ to $1 \times 10^{-5}$ using an exponential decay scheduling strategy with a decay factor of 0.999827. For the neural radiance field parameters in the low-resolution module, the learning rate is set from $1 \times 10^{-2}$ to $1 \times 10^{-4}$, using an exponential decay learning rate scheduling strategy with a decay coefficient of 0.999884. The neural radiance field parameters in the high-resolution module maintain a low learning rate of $0.0005(1 \times 10^{-2}/20)$. In the second stage, the low-resolution module remains unchanged, while the high-resolution module employs a smooth warm-up learning rate scheduling strategy, raising the learning rate to $0.003(0.3 \times 1 \times 10^{-2})$ and then decreasing it to $1 \times 10^{-4}$. Moreover, for the Lego and Hotdog scenes, some parameters are adjusted. The initial learning rate for the pose is set to $3 \times 10^{-3}$. The initial learning rate for the low-resolution neural radiance field parameters in the first phase is $5 \times 10^{-3}$, and the highest learning rate in the second stage for the high resolution is $9 \times 10^{-4}(0.3 \times 3 \times 10^{-3})$. The training was performed for 40,000 iterations for each scene.

**Table 1.** Parameters of the network structure.

|  | Parameter | Symbol | Value |
|---|---|---|---|
| Common parameters | Number of features Dimensions per entry | F | 2 |
|  | Hash table size | T | $2^{19}$ |
| Pose estimation module | Coarsest resolution | $N_{min}$ | 16 |
|  | Finest resolution | $N_{max}$ | 72 |
|  | Number of levels | $L_{res}$ | 3 |
| High-resolution reconstruction module | Coarsest resolution | $N_{min}$ | 128 |
|  | Finest resolution | $N_{max}$ | 4096 |
|  | Number of levels | $L_{res}$ | 16 |

The algorithm compared to ours is BAA-NGP. This algorithm currently offers the best scene reconstruction results using multi-resolution hash encoding in situations with inaccurate poses. Since the original experimental environment is different from ours, which may lead to variations in the results, we decided to evaluate the BAA-NGP algorithm under identical hardware and software environments and with the same perturbation parameters. The experimental details follow the settings in the BAA-NGP [33] paper. Specifically, we use the Adam optimizer and an exponential decay scheduling strategy. The learning rate for pose parameters changes from $1 \times 10^{-3}$ to $1 \times 10^{-5}$, and the learning rate for neural radiance field parameters changes from $1 \times 10^{-2}$ to $1 \times 10^{-4}$. Additionally, when processing the Materials and Ship data, we adjust the learning rate for pose parameters from $1 \times 10^{-2}$ to $1 \times 10^{-5}$ during training.

Furthermore, for a more comprehensive benchmark test, comparisons were made with the pose optimization parts of BARF [24] and NeRFAcc [32] in MLP-based NeRFs. Parameter settings were referenced from the literature, and the same initial conditions as in

this paper were set, specifically choosing a random seed of 3. The training was conducted for 200,000 iterations.

### 4.3.3. Results of BiResNeRF

The experimental results include both quantitative (Table 2) and qualitative (Figure 7) comparisons between our method and BAA-NGP. Through observations and statistics during the experiment, the effectiveness of the stage transition signals and the smooth warm-up learning rate scheduling strategy has been validated.

According to Table 2, our method performs well regarding pose estimation accuracy, rendering accuracy and training time. Among them, the average training time decreased by 34.96%, the average rotation error and translation error decreased by 34.66% and 39.73%, respectively, and the average rendering accuracy in terms of PSNR, SSIM and LPIPS increased by 4.90%, 0.96% and 18.14% respectively.

Based on Table 3, it can be seen that there is an improvement in rotation error, translation error and the quality of the rendered photos. In terms of training time, it is reduced by 94.58% compared to BARF and 79.28% compared to NeRFAcc, which is equivalent to an acceleration of $18.44\times$ and $4\times$, respectively.

**Table 2.** Quantitative comparison between BiResNeRF and the baseline BAA-NGP on Synthetic dataset. The translation error is the result of magnification by a factor of 100. The bold numbers in the table represent the superior results in the comparison of the two algorithms.

| Scene | Camera Pose Registration | | | | Visual Synthesis Quality | | | | | | Training Time(s) | |
| | Rotation(°)↓ | | Translation↓ | | PSNR↑ | | SSIM↑ | | LPIPS↓ | | | |
| | BAA | ours | BAA | ours | BAA | ours | BAA | ours | BAA | ours | BAA | ours |
|---|---|---|---|---|---|---|---|---|---|---|---|---|
| Lego | 0.038 | **0.038** | 0.121 | **0.107** | 33.704 | **34.871** | 0.977 | **0.983** | 0.023 | **0.017** | 1143.08 | **742.07** |
| Chair | **0.053** | 0.054 | **0.271** | 0.289 | 35.445 | **36.512** | 0.984 | **0.988** | 0.025 | **0.018** | 1464.40 | **804.51** |
| Drums | **0.028** | 0.031 | **0.098** | 0.169 | 25.160 | **25.502** | 0.921 | **0.925** | 0.090 | **0.089** | 1237.37 | **781.55** |
| Ficus | 0.030 | **0.029** | 0.136 | **0.102** | 31.307 | **31.789** | 0.977 | **0.979** | 0.034 | **0.028** | 1128.41 | **754.44** |
| Hotdog | 2.050 | **0.698** | 8.430 | **2.864** | 30.406 | **36.137** | 0.950 | **0.976** | 0.043 | **0.040** | 1150.72 | **766.78** |
| Materials | **0.038** | 0.149 | **0.134** | 1.414 | 28.638 | **28.944** | 0.943 | **0.952** | 0.082 | **0.065** | 1046.50 | **778.01** |
| Mic | 0.040 | **0.035** | 0.189 | **0.133** | 34.174 | **36.044** | 0.982 | **0.989** | 0.028 | **0.013** | 1312.24 | **819.45** |
| Ship | **0.065** | 0.496 | **0.282** | 0.745 | 30.052 | **31.286** | 0.888 | **0.904** | 0.114 | **0.089** | 987.59 | **768.65** |
| mean | 0.293 | **0.191** | 1.207 | **0.728** | 31.111 | **32.636** | 0.953 | **0.962** | 0.055 | **0.045** | 1183.79 | **776.93** |
| | | ↓34.66% | | ↓39.73% | | ↑4.90% | | ↑0.96% | | ↓18.14% | | ↓34.37% |

**Table 3.** Quantitative comparison of BiResNeRF with the baseline models BARF and NeRFAcc on Synthetic dataset. The translation error is the result of magnification by a factor of 100.

| Scene | Camera Pose Registration | | | | Visual Synthesis Quality | | | | | | Training Time(s) | |
| | Rotation(°)↓ | | Translation↓ | | PSNR↑ | | SSIM↑ | | LPIPS↓ | | | |
| | BARF | NeRFAcc | BARF | NeRFAcc | BARF | NeRFAcc | BARF | NeRFAcc | BARF | NeRFAcc | BARF | NeRFAcc |
|---|---|---|---|---|---|---|---|---|---|---|---|---|
| Lego | 0.077 | 0.059 | 0.257 | 0.215 | 28.377 | 30.020 | 0.929 | 0.948 | 0.048 | 0.038 | 9844 | 3647 |
| Chair | 0.095 | 0.074 | 0.396 | 0.334 | 31.202 | 32.361 | 0.955 | 0.965 | 0.044 | 0.039 | 9839 | 3440 |
| Drums | 0.051 | 0.038 | 0.201 | 0.191 | 23.914 | 24.772 | 0.900 | 0.917 | 0.099 | 0.079 | 12568 | 3532 |
| Ficus | 0.085 | 0.066 | 0.529 | 0.332 | 26.253 | 27.819 | 0.934 | 0.953 | 0.058 | 0.041 | 29349 | 3392 |
| Hotdog | 0.249 | 1.826 | 1.278 | 8.313 | 34.608 | 32.010 | 0.970 | 0.962 | 0.032 | 0.030 | 23485 | 3709 |
| Materials | 0.911 | 1.087 | 4.496 | 5.452 | 27.013 | 28.129 | 0.930 | 0.944 | 0.063 | 0.038 | 9839 | 3511 |
| Mic | 0.073 | 0.037 | 0.247 | 0.156 | 31.220 | 32.828 | 0.969 | 0.974 | 0.048 | 0.037 | 9803 | 3804 |
| Ship | 0.070 | 0.076 | 0.293 | 0.415 | 27.560 | 28.647 | 0.850 | 0.868 | 0.128 | 0.112 | 9907 | 4957 |
| mean | 0.202 | 0.408 | 0.962 | 1.926 | 28.768 | 29.573 | 0.930 | 0.941 | 0.065 | 0.052 | 14,329.250 | 3749.000 |
| | ↓5.11% | ↓53.13% | ↓24.37% | ↓62.21% | ↑13.44% | ↑10.35% | ↑3.48% | ↑2.17% | ↓31.06% | ↓13.06% | ↓94.58% | ↓79.28% |

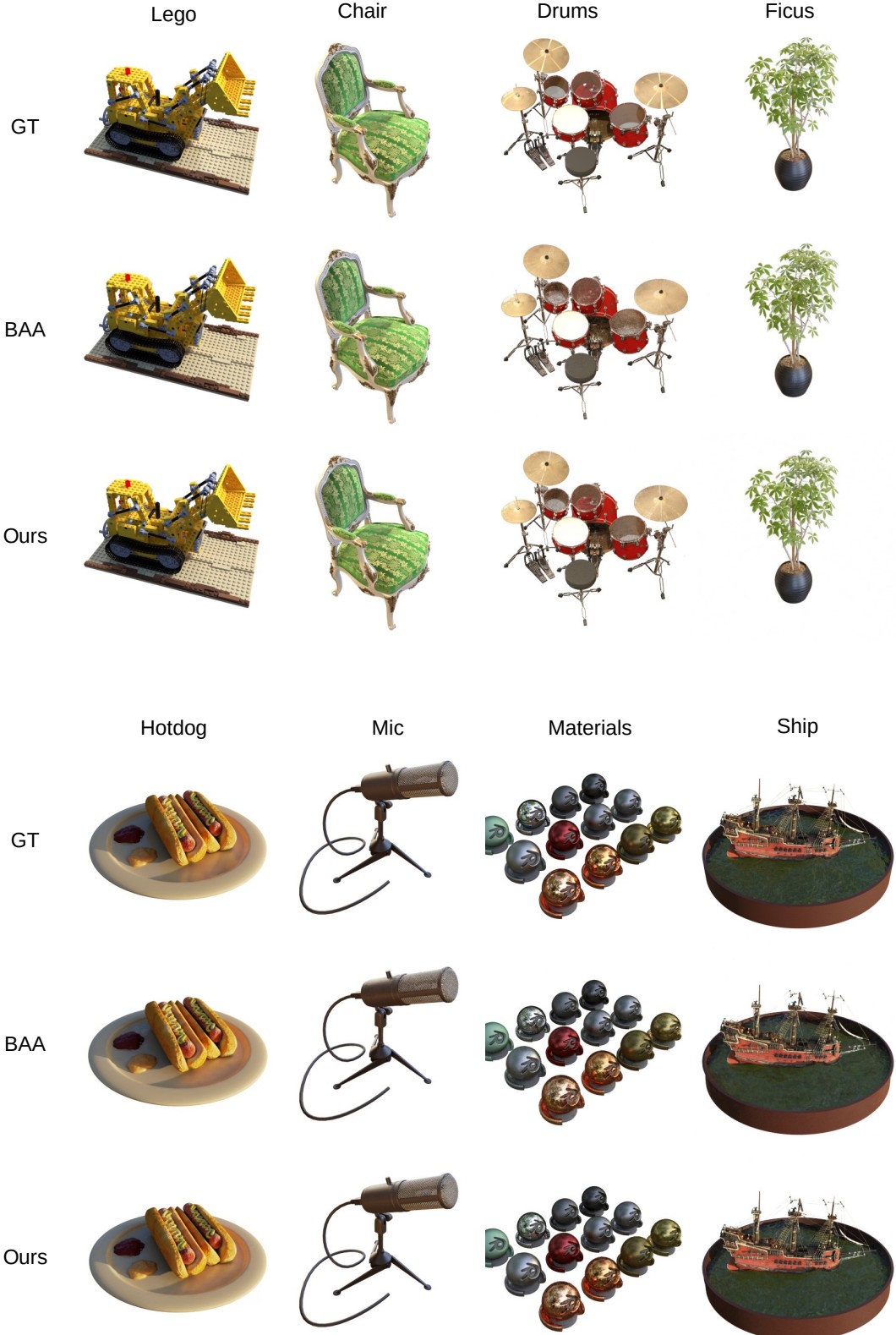

**Figure 7.** Qualitative results on Synthetic dataset. GT (Ground Truth) represents reference images, while BAA refers to images rendered by the method in [33].

### 4.3.4. Significance Analysis

Significance analysis helps us determine which evaluation indicators have statistically significant improvements among all evaluation criteria, thereby helping us identify significant progress in the aspects of our study.

In this experiment, significance analysis was conducted using the P-value as an indicator. A significance level of 0.05 was adopted. According to Table 4, in all tests, the P-value for training time was significantly lower than the 0.05 significance level, indicating that the improvements in training time made by the method presented in this paper were statistically significant. Furthermore, it was observed that other indicators were not significant. However, in terms of average values, the best levels were achieved in pose error and rendering quality. This indicates that the method proposed in this paper significantly reduced training time while ensuring optimal pose accuracy and rendering quality, thus meeting the expectations.

**Table 4.** Significance test results for different methods.

| Method | *p*-Value | | | | | |
|---|---|---|---|---|---|---|
| | Rotation | Translation | PSNR | SSIM | LPIPS | Training Time |
| BAA-NGP | 0.71 | 0.666 | 0.421 | 0.593 | 0.562 | $2.98 \times 10^{-6}$ |
| BARF | 0.942 | 0.713 | 0.055 | 0.095 | 0.225 | $1.98 \times 10^{-4}$ |
| NeRFAcc | 0.412 | 0.322 | 0.096 | 0.238 | 0.663 | $1.37 \times 10^{-10}$ |

### 4.3.5. Stability Analysis

The statistical data from Table 5 indicate that the average number of iterations in the first phase is 9.3 k, with an average time consumption of 160.85 s. The training time ranges from 1 min 21 s to 5 min 12 s, which shows that our method can adaptively adjust the number of iterations based on the scene's complexity. In the NeRF that utilizes a joint optimization pose framework, stability can be assessed through the loss value due to the absence of accurate pose information. As the training advances, the evident improvement in image quality and the decrease in loss values attest to the overall stability enhancement, indicating the gradual refinement of pose and three-dimensional structure. Consequently, the loss value not only corroborates the rendering improvement but also acts as an effective measure of pose stability.

**Table 5.** Signal triggering positions and times for stage transitions in different scenes.

| Scene | Lego | Chair | Drums | Ficus | Hotdog | Materials | Mic | Ship | Mean |
|---|---|---|---|---|---|---|---|---|---|
| Position(k) | 9 | 5.6 | 9.6 | 7.6 | 5 | 13.2 | 14.6 | 5.2 | 9.3 |
| Time | | 2 m 49 s | 1 m 26 s | 3 m 2 s | 2 m 18 s | 1 m 16 s | 4 m 12 s | 5 m 3 s | 1 m 21 s | 2 m 41 s |

We compare the smooth warm-up learning rate scheduling strategy with the cosine annealing learning rate scheduling strategy. By observing the magnitude of pose error changes in Figure 8a,b, it can be seen that the smooth warm-up learning rate scheduling strategy is noticeably more stable. Figure 8c shows the PSNR values of rendered pixels. Under the smooth warm-up learning rate scheduling strategy, the image accuracy improves more rapidly, while there is a cliff-like drop under the non-smooth strategy. This validates the effectiveness of the smooth warm-up learning rate scheduling strategy.

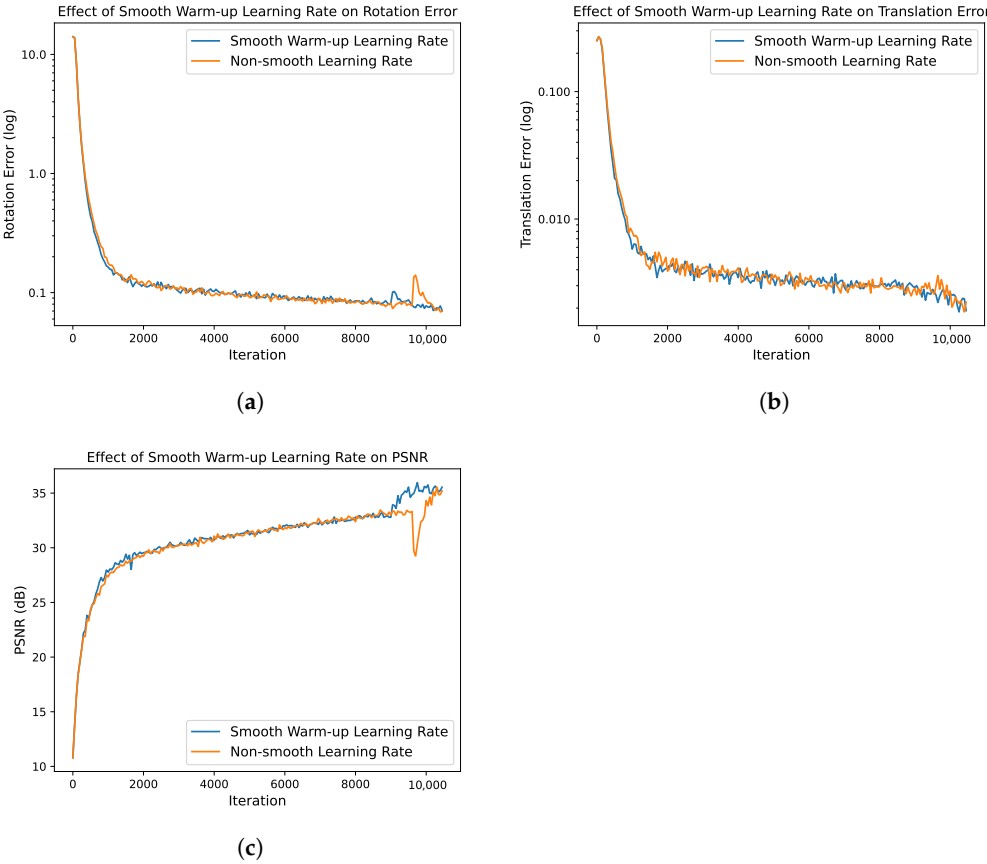

(**a**)

(**b**)

(**c**)

**Figure 8.** Performance difference between smooth warm-up learning rate scheduling strategy and non-smooth scheduling strategy in the scene of Lego.

## 4.4. Application in Low Textured Scenes

**Dataset**

We created a dataset through Blender. The dataset consists of a piece of ceramic. The surface of the ceramic is reflective and has low texture characteristics, making it impossible to obtain correct matching points between images. The object is placed at the coordinate origin, and the shooting path faces one side of the ceramic. There are a total of 17 images in this scene, all rendered at a resolution of [800, 800]. The experiments were conducted on data with the image size reduced to half of its original size [400, 400]. Some rendered images are shown in Figure 9.

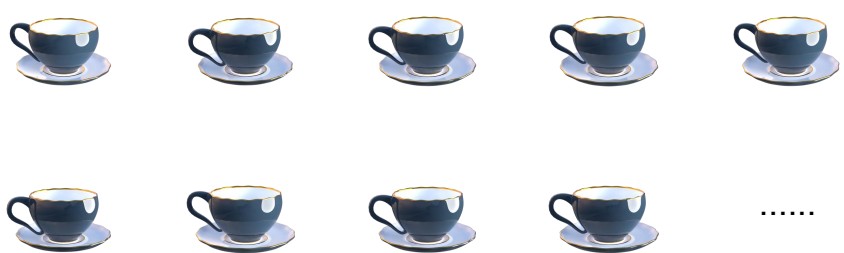

**Figure 9.** Partial data of a scene with low texture, reflective ceramic.

### 4.4.1. Experimental Setup

Since it is unreliable to use feature-based methods to obtain the pose information of objects in low-texture situations, perturbing the actual pose to simulate inaccurate pose conditions does not conform to reality. Therefore, in this experiment, we use scenes without pose data. In such cases, the reconstructed scenes can obtain estimated poses and render new viewpoint images. Then, by comparing the estimated poses with the real poses and

rendering new viewpoint images and depth images for display, we collectively verify the ability of our method to reconstruct scenes with challenging textures, demonstrating its great potential in research and application fields. The software and hardware environments used in the experiments are consistent with those in previous experiments.

### 4.4.2. Implementation Details

The experiment uses the BiResNeRF architecture, and the experimental details are basically the same as those in Section 4.3, only requiring modification of the learning rate parameters. For pose parameters, the learning rate decays from $1 \times 10^{-2}$ to $1 \times 10^{-5}$, with a decay coefficient of 0.999827. In the first stage, the learning rate of the low-resolution module is set from $5 \times 10^{-3}$ to $1 \times 10^{-4}$, with a decay coefficient of 0.999902. The learning rate of the high-resolution module remains low at $5 \times 10^{-5}$. In the second stage, the learning rate of the neural radiance field parameters of the high-resolution module increases to $0.005 \times 0.3$, and then decreases to 0.0001. Since there is no initial pose, all camera initial poses are aligned with the object, i.e., the coordinate origin, to ensure that the neural network can be trained normally. The training process was carried out for 40,000 iterations.

### 4.4.3. Results of Low-Texture Scene Experiments

As can be seen from Table 6, in the absence of a prior spatial distribution of camera poses, both translational and rotational errors are considerably large compared to the real poses. After aligning the estimated poses with the real poses, as observed in Figure 10, although there is a significant error in the poses of all cameras, the relative positions of all poses are correctly arranged. This one-to-one correspondence becomes more evident when the aligned results are projected onto the xy-plane in Figure 11. This indicates that the pose of each camera has been optimized. Furthermore, by observing the level of detail in the rendered images in Figure 12 and the state of the ceramics in the depth images, we can also infer that the structure of the object has been initially reconstructed. This validates the effectiveness of our algorithm in performing 3D reconstruction in scenes without pose, low textures and reflectiveness.

**Table 6.** Pose error in the ceramic scene.

| Scene | Rotation Error (°) | Translation Error |
|---|---|---|
| Ceramics | 25.848 | 1.416 |

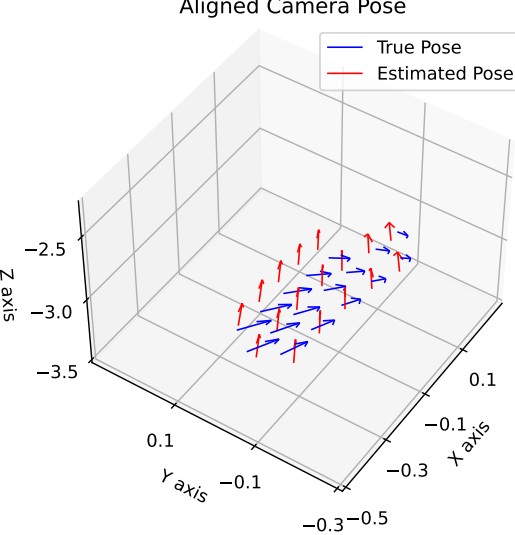

**Figure 10.** Visualization of aligned poses.

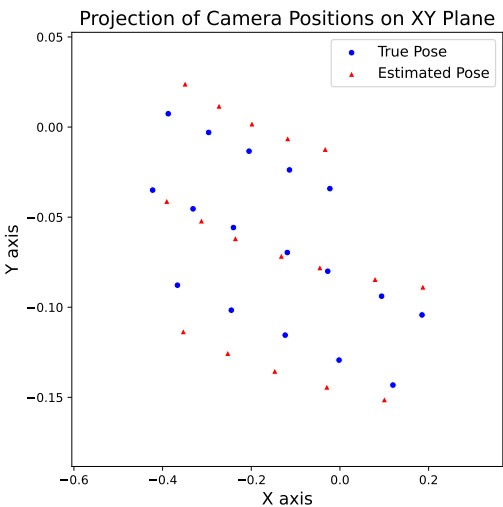

**Figure 11.** Projection of aligned poses onto the XY plane.

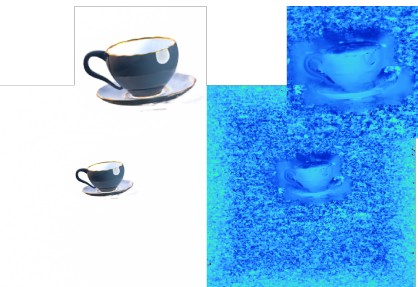

**Figure 12.** Rendering and depth map synthesized from the new perspective.

## 5. Discussion

In this section, we critically analyze our research findings, delve into the limitations and future work regarding the joint optimization method in NeRF, explore the stage transitions during the training process and discuss the potential impact of a research finding based on experimental conclusions and observations.

1. Although our method has achieved preliminary results in reconstructing inaccurate or pose-less, low-textured and reflective scenes, it still faces several challenges that are worthy of further research and improvement:

   - When the pose perturbation is too large, the joint pose optimization method still faces reconstruction failures.
   - In the complete absence of pose data, the spatial relationship between cameras cannot be too distant; otherwise, reconstruction is unachievable.
   - The method proposed in this paper improves both speed and accuracy; however, there is a slight increase in the occurrence of floaters. Minimizing the generation of floaters while maintaining efficient training is also a direction for further optimization.
   - Whether it is the BiResNeRF or the baseline method BAA-NGP, during evaluation, the convergence of poses exhibited varying degrees of instability, and an alternative learning rate was adopted for training in specific scenes. Therefore, to more conveniently apply joint pose optimization methods, further exploration is needed on the relationship between different parameters (such as the learning rate) and scene reconstruction.
   - The research has only been conducted on datasets generated from virtual scenes in Blender. Further research is needed on data acquisition methods, reconstruction methods and limitations in real-world scenarios.

2. The speed of error reduction is related to factors such as the learning rate of the neural radiative field and pose parameters and the number of sampling points on the ray, making it a highly complex issue. Although the current method can automatically perform stage transitions and ensure timely transitions after sufficient training in the first stage, the exact position of these transitions could be more precise. Therefore, determining the transition position more accurately remains a question worth further investigation.

3. We also discovered a research finding with potential in the experiments of this paper. Based on Table 3, during the first stage of training, when only utilizing the low-resolution pose estimation module, the average reconstruction time for scenes does not exceed 3 min. This means that, by using a three-layer low-resolution hash coding network, we can complete a preliminary reconstruction of the scene in about 3 min or even less and observe the results in real time. In contrast, BARF requires several hours to display the reconstruction results. This significant improvement in efficiency has brought tremendous momentum to related research in theory and application.

## 6. Conclusions

In conclusion, the BiResNeRF successfully achieved rapid and high-precision reconstruction for scenes with inaccurate poses. Initially, we introduced a feature fusion module, allowing hash encodings of different resolutions to participate more effectively in the reconstruction task. Following this, a two-stage training strategy was adopted, complemented by smooth warm-up learning rate scheduling and a coarse-to-fine sampling strategy, ensuring a smooth and expedited training process. As a result, through comparison with other algorithms, the reconstruction time was significantly reduced, with an average improvement of at least 34.37% on the synthetic dataset. Furthermore, in comparison to the MLP-based joint pose optimization method, this improvement was even more pronounced, ranging from 4 to 18.44 times. Importantly, this enhancement in speed did not compromise the accuracy of pose estimation or the quality of the rendered images. Finally, we also experimentally verified the adaptability and effectiveness of the algorithm in specific scenes without pose, low texture and reflection. BiResNeRF provides a new perspective on pose estimation based on low resolution, contributing to the ongoing discussions and research on joint pose optimization methods.

**Author Contributions:** Conceptualization, Z.G. and Q.X.; methodology, Z.G.; software, Z.G.; validation, Z.G., Q.X. and S.L.; formal analysis, Z.G.; investigation, Z.G.; resources, Q.X. and S.L.; data curation, Z.G.; writing—original draft preparation, Z.G.; writing—review and editing, X.X. and Z.G.; visualization, Z.G. and Q.X.; supervision, S.L. and X.X.; project administration, X.X.; funding acquisition, X.X. All authors have read and agreed to the published version of the manuscript.

**Funding:** This work was co-supported by Key Laboratory of Information and Computing Science Guizhou Province of Guizhou Normal University and Sichuan Provincial Cultural Relics and Archaeology Research Institute [grant numbers 11904-0621083].

**Institutional Review Board Statement:** Not applicable.

**Informed Consent Statement:** Not applicable.

**Data Availability Statement:** The Realistic Synthetic 360° data presented in this study are openly available in ACM Digital Library at https://doi.org/10.1145/3503250 (accessed on 26 October 2023), reference number 10.1145/3503250. The experimental data used to support the findings of this study are available from the corresponding author upon request. The data are not publicly available due to privacy.

**Acknowledgments:** The authors would like to show sincere thanks to those who have contributed to this research.

**Conflicts of Interest:** The authors declare no conflict of interest.

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
