# Peer review of "Bi-Resolution Hash Encoding in Neural Radiance Fields: A Method for Accelerated Pose Optimization and Enhanced Reconstruction Efficiency"

_applsci, doi:10.3390/app132413333_

Round 1

Reviewer 1 Report

Comments and Suggestions for Authors

I enjoyed reading the paper. The authors have presented an interesting manuscript.

Abstract, overview
The abstract is a concise description of the work. The introduction is well structured, and it covers all the concepts investigated in the methodological part. The previous work is well presented and integrated. I consider that this work brings added value in the field and the specific objectives of the manuscript are well related to the previous work developed in this domain.

Methodology
The research design used is appropriate in order to answer the research questions proposed by the authors. The methods are described properly. The results are clearly presented and are in relation to the concepts investigated.

Discussion and conclusions
The discussions are clear and concise. The conclusions are strongly related to the findings of the research work.

Format and style
All the format and style features were respected and are compliant with the requirements.

References
The format of the reference list fixes well to the specified format.

Plagiarism and any other ethical concerns about this study
I do not have any potential conflict of interest with regards to this paper.

Despite the good work done, there is still some room for improvement, as follows:

- How could the results here be utilized in other engineering sciences?

- I think some more literatures should be added. Besides the mentioned systems there are several others (like cost-effect BCI, eye-tracking, VR/AR) which are applied nowadays. It would improve the quality of the publication to a compare the non-invasive mobile EEG registration and the signal processing devices, mentioning other important human-computer interaction eye movement tracking would also improve quality, as such systems can be used in the analysis of programming technologies such as LINQ and algorithms or unconventional usage of different software tools, thus enabling, for example, cognition load or source code, algorithm description tools readability testing. In case of VR it would be welcomed to see an immersive virtual reality applications for supporting vision screening. It would also be elegant and the most important to address how such systems can change a person's self-confidence or self-efficacy in a way that several studies have analyzed the effect of software development course on programming on self-efficacy.

Comments on the Quality of English Language

The English language is good.

Reviewer 2 Report

Comments and Suggestions for Authors

BiResNeRF: A Method for Accelerated Pose Optimization and Enhanced Reconstruction Efficiency

This paper tackled the deployment issues of Neural Radiance Fields (NeRF) for object reconstruction of high fidelity. It proposes acceleration post estimation at high accuracy, by jointly optimizing both pose and scene reconstruction. Emphases of the proposed BiResNeRF were placed on strategically tunning and configuring baseline networks. Particularly, they were double resolution hash encoding, learning rates scheduling and sampling strategies. Its merits were elucidated on a synthetic dataset, consisting of a few generic 3D objects. The experiments indicated that BiResNeRF outperformed existing methods in terms of both accuracy and processing time. The paper is generally well written and structured. The problem statements and solutions are clear. The literature survey is concise, yet comprehensive. The experiments are technically sound. However, minor suggestions below could help strengthen its academic content and scientific quality.

Detailed Comments

1. The heading of section 2.2 reads confusing. Please revise.

2. Section 3.1: Please follow the standard notations of vector, scalar, and matrix for better clarity.

3. Section 3: Please ensure that all variables are defined right after their respective equations.

4. Section 3.2.1: It is not clear how the (a) low-res pose and (b) high-res scene reconstruction modules could resolve objects with challenging texture, other than efficient fusion. Please elaborate (schematic examples of synthetic pieces of small data would be helpful).

5. Section 3.2.2: Please provide reasons for choosing particular constant values (i.e., 0.8 and -0.1) for error trends characterization, and what were taken into account. Please provide some tests to ensure their robustness against different choices or guideline on selecting suitable ones.

5. Benchmarking against closely related works, e.g., [5-8], [9-10] or [16-19], could greatly help highlight the contribution. Currently, it was compared against [19] but only visually (Figure 7). Improvements over BAA-NGP methods (Table 2) are really marginal. In fact, significance analyses by P-Value would help confirm such improvement if any. In addition, as it employed generic public data, comparing the more objective metrices (between, most preferably, the recent studies) and detailed discussion should be feasible. Please consider.

6. Since the featured applications stated at the beginning include archaeological excavations and low-texture artifacts, please provide demonstrative experiments on those data in addition to the already included generic dataset. This is to ensure that BiResNeRF is indeed capable of reconstructing objects with challenging texture and feature extraction.

7. Please improve the graphic quality and clarity of all diagrams.

8. Please ensure that all graphs (e.g., Figure 5 and 6) are sufficiently large and their content (numbers and legends) are clear. Enlarge them if necessary.

9. Section 4.1.3: It is not clear whether all problems stated in the abstract (e.g., inaccurate or missing pose) are demonstrated and their solutions are shown in the experiments. Please confirm.

Comments on the Quality of English Language

generally good. minor polishing would help.

Author Response

I'm sorry for any inconvenience caused. The response letter I sent earlier was incorrect. Please consider this current version as the correct one.

Reviewer 3 Report

Comments and Suggestions for Authors

The article is devoted to a modern and relevant direction in neural networks - NeRF architecture, used for three-dimensional reconstruction of a scene based on a set of separate photographs. The relevance of research in this area aimed at improving accuracy and performance is high.

The article leaves a positive impression. There are several comments that do not reduce its scientific significance, but correcting them will improve the reading of the article for a wide range of researchers.

1. Line 42. It is worth giving an explanation of the abbreviations SFM, SLAM.

2. There is not enough mention in the text of the variable "I^" from formula 5.

3. In 196, a typo in the Raycam index

4. There is some comment to the figures, especially 3, 5, 6 and 8. I ask the authors to maximize their size, possibly due to vertical placement (the template also allows you to use the space on the left), because the text on them is extremely small to read.

5. I think it is worth clarifying and expanding the designations of paragraphs 4.1.3, 4.2.2 and 4.3.3 in order to remove the same names and add more specifics, taking into account which experiments these results relate to.

6. Please add an explanation to the abbreviation GT of Figure 7.

Round 2

Reviewer 1 Report

Comments and Suggestions for Authors

I enjoyed reading the paper. The authors have presented an interesting manuscript.

Abstract, overview
The abstract is a concise description of the work. The introduction is well structured, and it covers all the concepts investigated in the methodological part. The previous work is well presented and integrated. I consider that this work brings added value in the field and the specific objectives of the manuscript are well related to the previous work developed in this domain.

Methodology
The research design used is appropriate in order to answer the research questions proposed by the authors. The methods are described properly. The results are clearly presented and are in relation to the concepts investigated.

Discussion and conclusions
The discussions are clear and concise. The conclusions are strongly related to the findings of the research work.

Format and style
All the format and style features were respected and are compliant with the requirements.

References
The format of the reference list fixes well to the specified format.

Plagiarism and any other ethical concerns about this study
I do not have any potential conflict of interest with regards to this paper.

Despite the good work done, there is still some room for improvement, as follows:

- How could the results here be utilized in other engineering sciences?

- I think some more literatures should be added. Besides the mentioned systems there are several others (like cost-effect BCI, eye-tracking, VR/AR) which are applied nowadays. It would improve the quality of the publication to a compare the non-invasive mobile EEG registration and the signal processing devices, mentioning other important human-computer interaction eye movement tracking would also improve quality, as such systems can be used in the analysis of programming technologies such as LINQ and algorithms or unconventional usage of different software tools, thus enabling, for example, cognition load or source code, algorithm description tools readability testing. In case of VR it would be welcomed to see an immersive virtual reality applications for supporting vision screening. It would also be elegant and the most important to address how such systems can change a person's self-confidence or self-efficacy in a way that several studies have analyzed the effect of software development course on programming on self-efficacy.

Comments on the Quality of English Language

Minor editing of English language required.

Round 3

Reviewer 1 Report

Comments and Suggestions for Authors

- I think some more literatures should be added. Besides the mentioned systems there are several others (like cost-effect BCI, eye-tracking, VR/AR) which are applied nowadays. It would improve the quality of the publication to a the Comparison of the non-invasive mobile EEG registration and the signal processing devices, mentioning other important human-computer interaction eye movement tracking would also improve quality, as such systems can be used in the analyse the readability of LINQ code using an eye-tracking-based evaluation or an eye movement study in unconventional usage of different software tools, thus enabling, for example, cognition load or source code, algorithm description tools readability testing like in measuring cognition load using eye-tracking parameters based on algorithm description tools. In case of VR it would be welcomed to see an towards developing an immersive virtual reality applications for supporting vision screening – a user study. It would also be elegant and the most important to address how such systems can change a person's self-confidence or self-efficacy in a way that several studies have analyzed the effect of software development course on programming self-efficacy.

Author Response

Thank you for your guidance and comments.